# The Beneficial Effects of Traditional Chinese Exercises for Adults with Low Back Pain: A Meta-Analysis of Randomized Controlled Trials

**DOI:** 10.3390/medicina55050118

**Published:** 2019-04-29

**Authors:** Yanjie Zhang, Paul D. Loprinzi, Lin Yang, Jing Liu, Shijie Liu, Liye Zou

**Affiliations:** 1Lifestyle (Mind–Body Movement) Research Center, College of Sports Science, Shenzhen University, Shenzhen 518060, China; elite_zhangyj@163.com; 2Health and Exercise Science Laboratory, Institute of Sports Science, Seoul National University, Seoul 08826, Korea; 3Department of Health, Exercise Science and Recreation Management School of Applied Sciences, The University of Mississippi, Oxford, MS 36877, USA; pdloprin@olemiss.edu; 4Cancer Epidemiology and Prevention Research, Alberta Health Services, Calgary, AB T2S 3G3, Canada; lin.yang@ahs.ca; 5Departments of Oncology and Community Health Sciences, Cunning School of Medicine, University of Calgary, Calgary, AB T2N 4N1, Canada; 6Department of Martial Arts, Shanghai University of Sport, Shanghai 200438, China; liujing@sus.edu.cn; 7Department of Physical Education, Wuhan University of Technology, Wuhan 430070, China; liushijie0411@whut.edu.cn

**Keywords:** Tai Chi, mindfulness, Qigong, disability, randomized controlled trial

## Abstract

*Objective*: The aim of this meta-analytic review was to quantitatively examine the effects of traditional Chinese exercises (TCE) on pain intensity and back disability in individuals with low back pain (LBP). *Methods:* Potential articles were retrieved using seven electronic databases (Medline, Embase, Cinahl, Web of Science, Cochrane library, China National Knowledge Infrastructure, and Wanfang). The searched period was from inception to 1 March 2019. Randomized controlled trials (RCTs) assessing the effect of TCE on pain intensity and back disability in LBP patients were included. Pooled effect sizes were calculated using the random-effects models and 95% confidence interval (95% CI). *Results:* Data from eleven RCTs (886 individuals with LBP) meeting the inclusion criteria were extracted for meta-analysis. Compared with the control intervention, TCE induced significant improvements in the visual analogue scale (VAS) (*Hedge’s g* = −0.64, 95% CI −0.90 to −0.37, *p* < 0.001), Roland–Morris Disability Questionnaire (RMDQ) (*Hedge’s* g = −0.41, 95% CI −0.79 to −0.03, *p* = 0.03), Oswestry Disability Index (ODI) (*Hedge’s g* = −0.96, 95% CI −1.42 to −0.50, *p* < 0.001), and cognitive function (*Hedge’s g* = −0.62, 95% CI −0.85 to −0.39, *p* < 0.001). In a meta-regression analysis, age (β = 0.01, *p* = 0.02) and total exercise time (β = −0.0002, *p* = 0.01) were associated with changes in the VAS scores, respectively. Moderator analyses demonstrated that Tai Chi practice (*Hedge’s g* = −0.87, 95% CI −1.38 to −0.36, *p* < 0.001) and Qigong (*Hedge’s g* = −0.54, 95% CI −0.86 to −0.23, *p* < 0.001) reduced VAS scores. Interventions with a frequency of 1–2 times/week (*Hedge’s g* = −0.53, 95% CI −0.98 to −0.07, *p* = 0.02) and 3–4 times/week (*Hedge’s g* = −0.78, 95% CI −1.15 to −0.42, *p* < 0.001) were associated with reduced VAS scores, but this significant reduction on this outcome was not observed in the weekly training frequency of ≥5 times (*Hedge’s* g = −0.54, 95% CI −1.16 to 0.08, *p* = 0.09). *Conclusions:* TCE may have beneficial effects for reducing pain intensity for individuals with LBP, regardless of their pain status.

## 1. Introduction

Low back pain (LBP) usually occurs below the costal margin and above the inferior gluteal folds, across all adult ages, and often includes muscle tension, stiffness, and even sciatica [1]. Since the specific cause of LBP is poorly known, most individuals (approximately 85%) with LBP are marked as having nonspecific LBP (NLBP), which influences an individual’s activity of daily living and is becoming the leading cause of disability globally [2]. The 1-month prevalence of LBP is 23.2% worldwide [3], and the number of people with disability caused by LBP has increased by 54% in the last 30 years [2]. Although researchers and clinicians have used conventional medication and surgery to treat LBP for many years, a large proportion of patients continue to experience LBP without relief of significant pain [4,5].

As a cost-effective therapy, physical activity (i.e., walking, swimming, and resistance training) includes whole-body movements that do not only improve the back muscle strength but also improve body coordination and flexibility to help alleviate LBP [6,7]. Such a strategy results in small to large effect sizes in treating LBP [8,9,10]. For example, a meta-analysis showed that yoga reduced pain and increased physical function in LBP, with moderate to large effect sizes [8]. Furthermore, results from a systematic review concluded that exercise therapy had a small, yet positive effect size on improving LBP [10]. Cho and colleagues found that a 4-week core exercise program reduced LBP [11]. In general, these findings suggest that participating in multimodal interventions appear to relieve LBP.

Tai Chi and Qigong are Chinese health-promoting (lifestyle) exercises (traditional Chinese exercises; TCE), which are widely accepted as a multimodal exercise modality [12,13,14,15]. These are low-cost with mild to moderate exercise intensity and focus on physical–mental connection training [16,17,18]; meditative stage of mind and breathing control must be coordinated with slow body movement [19,20,21,22]. In recent decades, TCE has been effectively developed to treat LBP around the world. For example, Phattharasupharerk et al. found that Qigong exercise could significantly reduce pain intensity and back functional disability for office workers [23]. A recent randomized controlled trial also showed that individuals engaging in Tai Chi for 12 weeks could effectively achieve improvements on LBP [24]. Nevertheless, systematic review on the association between TCE and LBP is scarce. Only Li et al. have reported that Baduanjin Qigong reduced stiffness and facilitated the healing process in individuals with LBP [25]. 

However, early studies included small sample sizes and failed to provide ample evidence. Besides, no systematic review has been performed to analyze the latest studies on this topic. Therefore, the aim of this review was to examine the effects of TCE on LBP in terms of the visual analogue scale (VAS), Roland–Morris Disability Questionnaire (RMDQ), and Oswestry Disability Index (ODI), which are established assessments of LBP-related functionality.

## 2. Methods

### 2.1. Search Strategy

We searched seven electronic databases (Medline, Embase, Cinahl, Web of Science, Cochrane library, China National Knowledge Infrastructure, and Wanfang) from their inception to February 2019. The potential studies were retrieved by combining two sets of search terms, as follows: (1) “Tai Chi” or “Tai Chi Chuan” or “Qigong” or “Baduanjin” or “Yijinjing” or “Wuqinxi”; and (2) “Low back pain” or “Back pain”. In addition, additional publications were manually identified by searching the reference lists of related articles.

### 2.2. Inclusion and Exclusion Criteria

Only randomized controlled trials published in peer-review journals were included if they met the following inclusion criteria. (1) Participants diagnosed with LBP, with pain symptoms persisting for no less than 3 months. Furthermore, participants with LBP had to report pain that was confined to the lumbar vertebrae without severe organic or psychiatric disease. (2) At least one type of TCE (i.e., Tai Chi or Wuqinxi, Baduanjin, or Liuzijue) was used to treat LBP. (3) The control group was prescribed a non-TCE treatment, which could be classified as active treatment (i.e., strength exercise, back walking, or other physiotherapy) or passive control (i.e., waitlist, no treatment). (4) Pain intensity was the primary outcome measured by the VAS, with higher scores representing greater levels of pain. Secondary outcomes included the RMDQ and ODI, with higher scores indicating greater disability.

The exclusion criteria were: (1) participants had acute or sub-acute LBP, or LBP was caused by specific pathology or nerve root problems; (2) interventions included the combination of TCE with other interventions (i.e., acupuncture, core training); and (3) case-studies, cross-sectional studies, observational studies, or controlled study without randomization, or a review paper.

### 2.3. Data Extraction and Quality Assessment

One of investigators performed the electronic searches using a pre-created strategy. Titles and abstracts were independently screened by two investigators (Y.Z. and L.Y.Z.) after removing the duplicate and irrelevant publications. Subsequently, the remaining full-texts studies were checked. The third investigator (J.L.) was only invited to provide consensus to resolve any disagreements in the process of data extraction. The following information was extracted from each article: author and year of publication, characteristics of participants, sample size, intervention protocol, and outcome measures.

Methodological assessment was independently conducted by two authors (Y.Z. and Y.L.Z.) using the Physiotherapy Evidence Database (PEDro) scale [26]. This scale involves an 11-domain assessment: eligibility criteria (unscored), random allocation, concealed allocation, similar measures between groups at baseline, instructor blinding, assessor blinding, participant blinding, more than 85% retention rate, intention-to-treat analysis, between group statistical comparisons, and point estimates of at least one set of outcome data. One point was awarded for each item. From these scores, the studies were considered as excellent (9–10 points), good (6–8 points), fair (4–5 points), and poor (less than 4 points) quality.

### 2.4. Synthesized Analysis

The extracted outcome data from the original trials was expressed continuously with a comparison of pain-related outcomes between TCE and the control interventions. For each study, the effect size (ES) representing the difference in the outcome data between the two arms was calculated using the *Hedge’s g*. A random-effect model using inverse variance weights was robust enough to evaluate the ES due to the different study characteristics across the studies included. A positive ES indicated a more effective outcome for the experimental group. The magnitude of ES is usually classified as small (0.2–0.49), moderate (0.50–0.79), and large (≥0.8) [27]. Meanwhile, we examined the heterogeneity of ES using the I^2^ statistic and Q statistic. The three levels of heterogeneity were I^2^ < 25% (low heterogeneity), I^2^ < 50% (moderate heterogeneity), and I^2^ > 75% (high heterogeneity), respectively [27], of which low I^2^ indicates homogeneity across the trials and demonstrates the universality of the results. An Egger’s test was performed to examine publication bias, and publication bias was determined from a corresponding *p*-value less than 0.05.

Additionally, moderator analyses and meta-regression were conducted from categorical and continuous predictors. The categorical predictor consisted of: (1) the baseline pain intensity of participants, where the pain intensity was coded as 1–38 mm, 39–57 mm, and 58–100 mm, representing mild pain, moderate pain, and severe pain, respectively [28]; and (2) the characteristics of the control group and experimental group, where the control groups were categorized as active interventions or passive interventions, and the mode of experimental intervention was classified as Tai Chi exercise or Qigong exercise. The coding of the exercise session time was defined as ≤45 min and >45 min. The duration of exercise was coded as short (≤ 12 weeks) or medium (13–26 weeks). The weekly frequency was coded as low (≤2), medium (3–4), or large (≥5). The continuous predictors included the mean age of the subjects and the total time of TCE exercise. All analyses were conducted using the Comprehensive Meta-Analysis Software.

## 3. Results

### 3.1. Search Results

A total of 319 citations were initially identified from the electronic database (Figure 1). Thirty-nine potential studies were retained after screening the titles and abstracts and removing the duplicate publications (*n* = 139). Furthermore, 39 relevant studies were evaluated through reading full-texts, and 11 RCT studies [23,24,29,30,31,32,33,34,35,36,37] were included in this meta-analysis.

### 3.2. Study Characteristics and Quality

Detailed characteristics of the 11 included studies are described in Table 1 and were published in the last decade (from 2011 to 2019). Six studies were published in Chinese-language peer review journals [31,32,34,35,36,37]. All participants (*n* = 886) with a mean age from 35 to 74 years were suffering from mild (1 study) [37], moderate (6 studies) [23,29,30,31,33,34], and severe (four studies) [24,32,35,36] LBP, respectively. The sample size of each study ranged from 15 to 176. In terms of study design, the TCE program was used to treat LBP in all experimental groups, where four studies were conducted using Tai Chi and seven studies were conducted using Qigong. Either active interventions (i.e., core training) or passive interventions (i.e., waitlist) were employed in the control group. The duration of TCE varied from two weeks to 24 weeks. The weekly frequency of TCE ranged from one to seven times, and each session time lasted 20 to 90 min. With regard to the outcomes, VAS was assessed in 10 studies [23,24,29,30,31,32,33,34,35,36], RMDQ was assessed in four studies [23,29,30,35], and five studies [32,34,36,37] evaluated the ODI. Only one study [33] reported on suspected side effects such as dizziness (*n* = 12) and increased pain (*n* = 2). The remaining studies did not report any exercise-related side effects. 

Details of the quality assessment for each study using the PEDro scale are summarized in Table 2. Overall, the quality of all included studies was good. The randomization sequence was computer generated in all studies, and five studies described the process of concealed allocation with sufficient detail. The majority of studies did not use subject, therapist, and assessor blinding. The less than 15% loss rate, intention-to-treat analysis, and the between-group statistical comparison for more than one outcome were described in all included studies.

### 3.3. Synthesis of Results

#### 3.3.1. Effect of TCE on Pain Intensity

Pooled analysis of the effect of TCE on pain intensity found significant heterogeneity between studies (*I^2^* = 74.7%). From the 13 trials [23,24,29,30,31,32,33,34,35,36], and using a random-effect model, TCE was effective in reducing pain intensity (*Hedge’s g* = −0.64, 95% CI (confidence interval) −0.90 to −0.37, *p* < 0.001) (Figure 2). This means that *Hedge’s g* is considered a moderate ES.

#### 3.3.2. Effects of MBE on RMDQ and ODI

As shown in Table 3, the pooled analysis from four trials [23,29,30,35] showed that TCE was significantly associated with an improvement in the RMDQ (95% CI −0.79 to −0.03). The mean *Hedge’s g* was −0.41, *p* = 0.03, which is considered a small ES. Regarding ODI, the aggregate result revealed a significant favorable effect on ODI (95% CI −1.42 to −0.50). The mean *Hedge’s g* was −0.96, *p* < 0.001, which is considered a large ES.

#### 3.3.3. Moderator Analysis

Moderator analysis using separate models was employed to examine potential sources of variance. All results are presented in Table 4.

*Study design moderators:* In terms of the control group, two types of controls (active intervention and passive intervention) were employed in the original studies. Seven studies employed an active intervention (i.e., stretching, core training). The type of control group did contribute to statistically significant differences for the ES estimate (*Q* = 4.50, *p* = 0.03). TCE had a significant improvement on VAS (*Hedge’s g* = −0.40, 95% CI −0.64 to −0.15, *p* = 0.001; *Hedge’s g* = −0.99, 95% CI −1.48 to −0.50, *p* < 0.01) compared with active interventions and passive interventions, respectively. Notably, allocation concealment produced a significant difference on the VAS (*Q* = 8.73, *p* = 0.01). Using an appropriate allocation concealment had a small and significant ES on VAS (*Hedge’s g* = −0.34, 95% CI −0.57 to −0.12, *p* < 0.01). In contrast, a large and significant ES on VAS was found in favor of non-allocation concealment (*Hedge’s g* = −1.04, 95% CI −1.45 to −0.64, *p* < 0.01). In addition, when the baseline pain intensity was evaluated as a categorical moderator, there was no significant difference in the ES (*Q* = 0.40, *p* = 0.53): moderate pain intensity (*Hedge’s g* = −0.57, 95% CI −0.90 to −0.25, *p* < 0.01) and severe pain intensity (*Hedge’s g* = −0.77, 95% CI −1.28 to −0.26, *p* < 0.01).

*TCE moderators:* For interventions in the experimental group, TCE included Tai Chi (eight arms) and Qigong (three arms) in our current meta-analysis. The statistically significant difference of the ES evaluated was not observed (*Q* = 1.15, *p* = 0.28) in the type of experimental group (between TC exercise and Qigong exercise), of which we further observed the effect of TC (*Hedge’s g* = −0.87, 95% CI −1.38 to −0.36) or Qigong (*Hedge’s g* = −0.54, 95% CI −0.86 to −0.23) on the VAS. Similarly, the frequency of TCE intervention did not produce a statistically significant difference across the three levels (*Q* = 0.90, *p* = 0.64). There was no statistically significant difference for exercise session time (*Q* = 0.10, *p* = 0.75). A moderate and significant reduction in the ES was attributed to the exercise session time (≤45 min, *Hedge’s g* = −0.64, 95% CI 0.46 to 0.82, *p* < 0.01) when compared with longer duration exercise (>45 min), which contributed to a small ES (*Hedge’s g* = −0.73, 95% CI −1.26 to −0.19, *p* < 0.01).

#### 3.3.4. Meta-Regression

Regarding the effects of TCE on VAS, both age (*β* = 0.01036, *p* = 0.02) and total exercise time (*β* = −0.00020, *p* = 0.01) influenced the ES, implying that the effect of TCE on the VAS decreased with advancing age, and was still significantly effective compared with other interventions, additionally, practicing TCE for a longer time may significantly reduce VAS. However, we found no significant relationship between pain intensity at the baseline and TCE (*β* = −0.00016, *p* = 0.99).

#### 3.3.5. Publication Bias

The publication bias was assessed using Egger’s test. The results showed a *p*-value of 0.11, greater than 0.05, which reflected no publication bias (from a statistical significance perspective) for this present meta-analysis. Due to the small number of trials included, publication bias of other outcomes was not performed. 

## 4. Discussion

This current meta-analysis is the first, to our knowledge, to evaluate the effect of TCE on LBP patients. Our findings suggest that TCE is an effective therapy and may influence VAS, RMDQ, and ODI. Improvement in pain intensity was consistently observed in two types of TCE, and no adverse events occurred among the included studies. The efficacy of TCE may be attributed, in part, to combining substantial components of the American College of Sports Medicine (ACSM) recommendations [38] such as muscle strength, flexibility, and stretching training.

Pain intensity refers to how much an individual is hurt by their LBP and can be quantified to estimate the magnitude of severity. Two scales, VAS and NRS, are most commonly used to evaluate pain intensity using a numeric scale (0–100 mm) in LBP [39]. Results from this meta-analysis indicate that TCE can significantly reduce the VAS scores for LBP patients. The ES for VAS was of a moderate effect (0.64). With respect to MRDQ and ODI, both measurement tools are recommended by an internal expert panel to assess the LBP disability [39]. The MRDQ is usually employed to examine the limited range of physical function (i.e., walking, sitting, bending over, dressing, and so on). The pooled result from four trials revealed that after practicing TCE, LBP patients significantly improved their MRDQ scores. The calculated ES was 0.41, which is considered a small effect. After practicing TCE, patients achieved significant improvements in their performance in ODI.

As heterogeneity across studies is common in meta-analyses [40], it is not surprising that there was considerable heterogeneity on the effect of TCE on the VAS. Meta-regression showed that age and total exercise time were moderators of the effect of TCE on VAS, which implies that the effect of TCE on pain intensity decreased with age, but long-term exercise may significantly reduce pain intensity. In addition, the study design characteristics were also explored as potential moderating variables. First, pain intensity at the baseline did not change the results of the meta-analysis. Part of this may be that practicing TCE, like other exercise forms, can relieve patient pain through the use of various poses and movements that enhance back stabilizer muscles [41]. Second, regarding the type of control group, significant ES was observed for either the active or passive controls. This implies that other interventions (i.e., core training, stretching, usual care) were less effective than TCE in reducing pain intensity, but at least the active intervention (i.e., core training, stretching) was more effective than passive controls (i.e., usual care, waitlist). For allocation concealment, seven trials using adequate allocation concealment were considered to be of good methodological quality (Table 2). Moderator analysis suggested that a significant difference in the VAS between allocation concealment and non-allocation concealment was observed. This suggests that studies using insufficient allocation concealment achieved a large ES in our meta-analysis. It is possible that researchers may unconsciously or consciously influence the allocation when participants are assigned to the intervention groups, which may have exaggerated the intervention effects, as some individuals may be more likely to respond favorably to TCE.

Furthermore, exercise moderators involving TCE modality, frequency, and exercise session time are crucial to investigate the effects of TCE prescription on changes in pain intensity. Similar efficacies in improving pain intensity were observed in both Tai Chi and Qigong modalities. The moderator analysis indicated that each session time for ≤45 min or >45 min and frequency (one to four times) could significantly contribute to moderate ESs on pain intensity. However, our current meta-analysis showed that although the ES was 0.57, practicing TCE more than four times per week did not significantly benefit pain intensity (*p* = 0.10). The reason for this finding is not fully understood. Considering the duration of TCE was mostly short-term, duration-based moderator analyses were not performed. Although the mechanism of TCE for LBP is still unclear, some possible mechanisms may be explained according to the characteristics of TCE. TCE not only focuses on the regulation of internal energy, but also on the improvement of physical function through meditation and breathing, posture control as well as strength and flexibility training [42]. First, meditation and rhythmic breathing, as the foundation of mind–body exercise, can effectively elevate vitality and its flow in the body to reduce pain and stiffness [43]. A recent meta-analysis demonstrated that reduced muscular pain for adults with chronic diseases could be attributed to mindfulness-based training [44]. Moreover, a prior study suggested that practicing Tai Chi for three months could significantly reduce LBP in adults aged 50 years and older [24]. These findings suggest that TCE can contribute to reducing pain intensity for individuals suffering from LBP. Second, TCE incorporates muscular strength, stabilization, static, and dynamic balance to reduce pain, and these principles are very similar to other exercises (i.e., as core stabilization programs) in modulating pain intensity and physical function [29,45]. In particular, practitioners need to transfer momentum of the upper body and low limbs by integrating slow coordinating postures, and utilize lower extremity strength and balance when they practice TCE. It is likely that TCE can synergistically enhance the lumbar muscle and decrease back pain through improving the lumbar extensor muscles and complementary core-hip muscles. Additionally, one of the characteristics of TCE is flexibility, and improved lumbar flexibility is associated with reduced LBP [46]. While practicing TCE, postural training can improve flexibility depending on the stimulation of muscles during static and dynamic movements. Furthermore, holistic TCE can improve the back range of motion through posture control, lumbar muscular flexibility, core strengthening, and breathing. Thus, TCE appears to contribute to the improvement of pain intensity and back disability and there are various potential multilevel mechanisms to explain this effect.

This meta-analysis has several limitations. First, although we used seven electronic databases to identify potential studies, it is likely that some studies were not retrieved because they were not indexed in these databases. Second, studies using a sole intervention (Tai Chi or Qigong) in the experiment group were included. Several studies combining TCE and other interventions (i.e., acupuncture, strength training) were excluded because the aim of this present meta-analysis was to investigate the effect of TCE on LBP without considering other factors. Third, most studies did not use a blinding strategy (i.e., participant blinding, assessor blinding), which could have produced subjective expectation bias and exaggerated the research results; however, this was not a limitation of our meta-analysis, but rather, a general limitation of the studies conducted on this topic. Finally, the duration of TCE was mostly short-term (less than 12 weeks) among the included studies, and thus, it is unclear whether the long-term practice of TCE is beneficial for LBP patients.

## 5. Conclusions

The results of the current meta-analysis demonstrate that TCE may have a positive effect on modulating pain intensity, MDRQ, and ODI for people with LBP. To better understand the effects of TCE on LBP patients, rigorous study designs are necessary. Future studies should be consistent with the consolidated standards of reporting trials (CONSORT) statement.

## Figures and Tables

**Figure 1 medicina-55-00118-f001:**
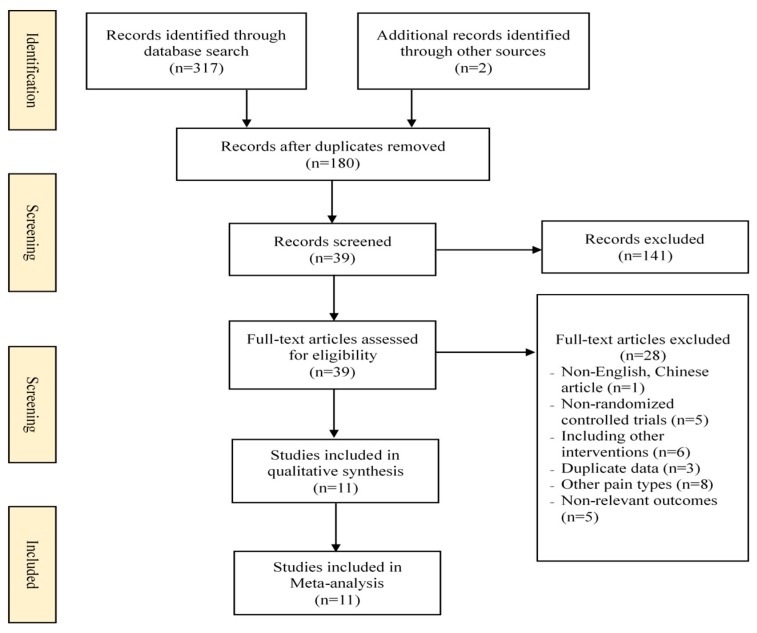
Flowchart of the study selection.

**Figure 2 medicina-55-00118-f002:**
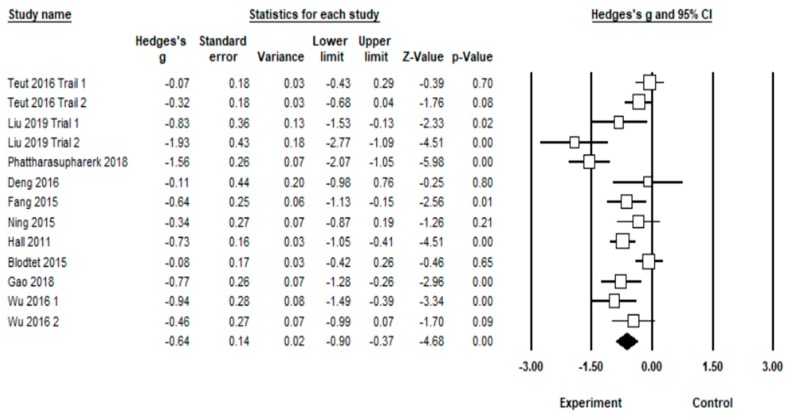
Effect of traditional Chinese exercises (TCE) on the visual analogue scale (VAS).

**Table 1 medicina-55-00118-t001:** Characteristics of randomized controlled trials in the meta-analysis.

Study	Participants			Interventions			Outcomes Measured	Safety
Publication Year	Status	Sample size	Age (years)	Experiment	Control	Duration	Primary and/or	Adverse
	(PI)	(female, male)					secondary outcome	effect
Phattharasupharerket al., (2018) [23]	LBP (50)	72	35.2	1 × 60 min/week + daily practice Qigong	Waitlist	6 weeks	Pain intensity (VAS),	No
		E = 36; C = 36					Low back pain disability (RMDQ)	
Liu et al., (2019) [24]	LBP (57)	43	74	3 × 60 min/week	C1: Core training	12 weeks	Pain intensity (VAS)	No
		E = 15; C1 = 15; C2 = 13		Tai Chi	C2: no intervention			
Hall et al., (2011) [29]	LBP (50)	160	44	2 × 40 min/week	Wait-list	10 weeks	Pain intensity (NRS),	No
		E = 80; C = 80		Tai Chi			Low back pain disability (RMDQ)	
Blödt et al., (2015) [30]	LBP (56)	127	46.7	1 × 90 min/week,	1 × 60 min/week	12 weeks	Pain intensity (VAS),	Dizziness (*n* = 12)
		E = 64; C = 63		Qigong	Strengthening		Low back pain disability (RMDQ	Increased pain (*n* = 2)
Ning et al., (2015) [31]	LBP (58)	52	41.4	3 × 30 min/week,	3 × 30 min/week	12 weeks	Pain intensity (VAS),	No
		E = 26; C = 26		Wuqinxi	Core training		Low back pain disability (ODI)	
Fang et al., (2015) [32]	LBP (52)	63	53.4	3–4 × 45 min/week,	3–4 × 45 min/week,	24 weeks	Pain intensity (VAS)	No
		E = 32; C = 31		Wuqinxi	McKenzie training			
Deng (2016) [33]	LBP (70)	15	50.4	2 × 20 min/day	1 × 20 min/day	2 weeks	Pain intensity (VAS),	No
		E = 8; C = 7		Yijinjing	Acupuncture		Low back pain disability (MRMQ)	
Teut et al., (2016) [34]	LBP (51)	176	73	1 × 90 min/week,	C1: 2 × 45 min/week	12 weeks	Pain intensity (VAS)	No
		E = 58; C1 = 61; C2 = 57		Qigong	Yoga; C2: Waitlist			
Wu 2016 [35]	LBP 53	78	39	3 × 30–35 min/week,	C1: 3 × 30–40 min/week (SB)	12 weeks	Pain intensity (VAS),	No
		E = 26; C1 = 26; C2 = 26		Wuqinxi	C2: 3 × 30–40 min/week (CT)		Low back pain disability (ODI)	
Gao et al., (2018) [36]	LBP (76)	60	36	2 × 30 min/day,	Usual care	8 weeks	Pain intensity (VAS),	No
		E = 30; C = 30		Baduanjin			Low back pain disability (ODI)	
Liu et al., (2018) [37]	LBP (31)	40	57	3 × 60 min/week	C1: Core training	12 weeks	Low back pain disability (ODI)	No
		E = 14; C1 = 13; C2 = 13		Tai Chi	C2: no intervention			

Note: PI = pain intensity; LBP = low back pain; E = Experiment; C = Control; RMDQ = Roland–Morris Disability Questionnaire; VAS = Visual Analog Scale; SB = Swallowing balance; CT = core training.

**Table 2 medicina-55-00118-t002:** Methodological quality of the included studies (PEDro assessment).

Study	Score	Methodological Quality	PEDro Item Number										
			1	2	3	4	5	6	7	8	9	10	11
Phattharasupharerk et al. 2018 [23]	8	Good	✓	✓	✓	✓				✓	✓	✓	✓
Liu et al. 2019 [24]	7	Good	✓	✓		✓				✓	✓	✓	✓
Hall et al. 2011 [29]	8	Good	✓	✓	✓	✓				✓	✓	✓	✓
Blödt et al. 2015 [30]	8	Good	✓	✓	✓	✓				✓	✓	✓	✓
Ning et al. 2015 [31]	8	Good	✓	✓	✓	✓				✓	✓	✓	✓
Fang et al. 2015 [32]	8	Good	✓	✓	✓	✓			✓	✓	✓	✓	✓
Deng 2016 [33]	7	Good	✓	✓		✓				✓	✓	✓	✓
Teut et al. 2016 [34]	8	Good	✓	✓	✓	✓				✓	✓	✓	✓
Wu 2016 [35]	8	Good	✓	✓		✓			✓	✓	✓	✓	✓
Gao et al. 2018 [36]	7	Good	✓	✓		✓				✓	✓	✓	✓
Liu et al. 2018 [37]	7	Good	✓	✓		✓				✓	✓	✓	✓
Studies were classified as having excellent (9–10), good (6–8), fair (4–5), or poor (<4).													

Scale of item score: ✓, present. The Physiotherapy Evidence Databa (PEDro) scale criteria are: (1) eligibility criteria; (2) random allocation; (3) concealed allocation; (4) similarity at baseline on key measures; (5) subject blinding; (6) therapist blinding; (7) assessor blinding; (8) more than 85% follow-up of at least 1 key outcome; (9) intention-to-treat analysis; (10) between-group statistical comparison for at least 1 key outcome; and (11) point estimates and measures of variability provided for at least 1 key outcome.

**Table 3 medicina-55-00118-t003:** The results for the effects of TCE vs. the control intervention.

Outcomes	Number of Trials	Meta-Analysis			Heterogeneity		
		*He* *dges’g*	95% CI	*p*-Value	*I*^2^ %	*Q*-Value	df(*Q*)
RMDQ	4	−0.41	−0.79 to −0.03	0.03	66.1%	8.86	3
ODI	5	−0.96	−1.42 to −0.50	0.00	67.5%	12.32	4

**Table 4 medicina-55-00118-t004:** Moderator analysis for TCE versus the control group.

**Categorical Moderator**	**Outcome**	**Level**	**No. of Studies/** **Comparisons**	**Hedges’ g**	**95% Confidence Interval**	***I*** **^2^** **, %**	**Test for Between-Group** **Heterogeneity**		
							***Q*** **-Value**	**df(*Q)***	***p*** **-Value**
Study design moderators									
Control Type	VAS	Active	8	−0.40	−0.64 to −0.15	45.2 %	4.50	1	0.03
		Passive	5	−0.99	−1.48 to −0.50	82.3%			
Allocation Concealment	VAS	Yes	7	−0.34	−0.57 to −0.12	65.2%	8.73	1	0.01
		No	6	−1.04	−1.45 to −0.64	49.7%			
Baseline pain intensity	VAS	Moderate	8	−0.57	−0.90 to −0.25	79.2%	0.40	1	0.53
		Severe	5	−0.77	−1.28 to −0.26	66.7%			
Exercise moderators									
Exercise Type	VAS	TC	9	−0.87	−1.38 to −0.36	69.9%	1.15	1	0.28
		Qigong	4	−0.54	−0.86 to −0.23	75.9%			
Frequency	VAS	1–2	5	−0.53	−0.98 to −0.07	86.8%			
		3–4	6	−0.78	−1.15 to −0.42	57.3%	0.90	2	0.64
		≥5	2	−0.54	−1.16 to 0.08	39.6%			
Exercise session time	VAS	≤45	7	−0.64	−0.82 to −0.46	0%	0.10	1	0.75
		>45	6	−0.73	−1.26 to −0.19	87.6%			
**Continuous Moderator**	**Outcome**	**No. of Studies/** **Comparisons**	β	**95% Confidence Interval**		***Q*** **-Value**		**df(*Q*)**	***p*** **-Value**
Age	VAS	13	0.01036	0.00163 to 0.01909		5.41		1	*0.02*
Total exercise of time	VAS	13	−0.00020	−0.00036 to −0.00005		6.79		1	0.01
Pain intensity of participants	VAS	13	−0.00016	−0.01902 to 0.01871		0.01		1	0.99

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
