# Peer review of "The Beneficial Effects of Traditional Chinese Exercises for Adults with Low Back Pain: A Meta-Analysis of Randomized Controlled Trials"

_medicina, 2019, doi:10.3390/medicina55050118_

Round 1
Reviewer 1 Report
This is a very interesting and useful systematic review that reviews and meta-analyzes the clinical evidence of the effects of Traditional Chinese Exercises (TCE) such as tai chi and qigong on lower back pain. Similar to other important reviews by the same group of authors, this review (1) focus on the clinical studies among patient population only, (2) set a higher standard for inclusion so that only high-quality RCTs with active control would be included in the review; and (3) evaluate both pain and disability outcomes. This reviewer can feel the amount of passion and efforts the authors have put into this special and complicated report, and sense the difficulties applying the western medicine standard to evaluate the clinical applications of an eastern traditional practice that was not meant to be used as a clinical therapy. The review is in general in good shape, although I would like to see some clarification and fine tune before its publication. I can probably give a lot of appraisal for its strength, significance, and uniqueness in the field; however, I think some critiques, comments or questions may help this review to become more useful. Here are my comments and questions on the problems in the current version of review:
1) Authors mentioned that they searched 7 databases for inclusion, two of them are Chinese databases (WanFang and CNKI), but did not state if they found anything published in Chinese, and what they did with the Chinese publications. I would like to see a clarification on this, and why we did not see any Chinese publications in this report.
2) The definition of "Traditional Chinese Exercise" (TCE) is a relatively new term, and did not well defined in the study. Why does it include taiji and qigong only, but not include other forms such as reiki, mindfulness, meditation, yoga, tui-shou, etc. the definition of therapy is an important first step before such a systematic review.
3) Among the excluded studies, 5 of them were due to non-relevant outcomes. I wonder what kind of outcomes they are, since systematic review is supposed to offer an overview summary of the field, if 5 studies had outcomes that reflect the general concern from physicians and patients, we may consider include them into the review instead of excluding them due to different outcomes....
4) The paper may benefit from a through English proofread as I have seen a few English errors,.
5) Publication bias. The author consider the result of Egger's test p = 0.11 reflected no publication bias, which may not be correct, since p=0.11 reflect a high probability (89%) to imply some publication bias, but has not reach statistically significance level.
6) Lifestyle issue. Both Qigong and Taichi is not designed as a clinical therapy, but a way of life in traditional Chinese medicine. The reviewed studies took TCE as a clinical therapy and show some significant clinical outcome. That is great! However, Taichi and Qigong is designed to generate a different lifestyle, which should affect the practitioner's behavior pattern and life-style in a long term. Future studies should keep that in mind, and include more quality of life and lifestyle outcomes in assessment.
Author Response
Reviewer 1
This is a very interesting and useful systematic review that reviews and meta-analyzes the clinical evidence of the effects of Traditional Chinese Exercises (TCE) such as tai chi and qigong on lower back pain. Similar to other important reviews by the same group of authors, this review (1) focus on the clinical studies among patient population only, (2) set a higher standard for inclusion so that only high-quality RCTs with active control would be included in the review; and (3) evaluate both pain and disability outcomes. This reviewer can feel the amount of passion and efforts the authors have put into this special and complicated report, and sense the difficulties applying the western medicine standard to evaluate the clinical applications of an eastern traditional practice that was not meant to be used as a clinical therapy. The review is in general in good shape, although I would like to see some clarification and fine tune before its publication. I can probably give a lot of appraisal for its strength, significance, and uniqueness in the field; however, I think some critiques, comments or questions may help this review to become more useful. Here are my comments and questions on the problems in the current version of review:
1) Authors mentioned that they searched 7 databases for inclusion, two of them are Chinese databases (WanFang and CNKI), but did not state if they found anything published in Chinese, and what they did with the Chinese publications. I would like to see a clarification on this, and why we did not see any Chinese publications in this report.
Response: As suggested, we have made clarification within subsection 3.2 of “study characteristics and quality” as follows: Six studies were published in Chinese-language peer review journals [31, 32, 34-37].
2) The definition of "Traditional Chinese Exercise" (TCE) is a relatively new term, and did not well defined in the study. Why does it include taiji and qigong only, but not include other forms such as reiki, mindfulness, meditation, yoga, tui-shou, etc. the definition of therapy is an important first step before such a systematic review.
3) Response: Thank for your suggestion. Tai Chi and Qigong are the most popular Chinese health-promoting exercises and they have shared similar traditional Chinese medicine theory but are different from reiki, mindfulness, meditation, yoga, etc. In addition, as suggested, we have redefined “these two exercises” as Chinese health-promoting (lifestyle) exercises.
4) Among the excluded studies, 5 of them were due to non-relevant outcomes. I wonder what kind of outcomes they are, since systematic review is supposed to offer an overview summary of the field, if 5 studies had outcomes that reflect the general concern from physicians and patients, we may consider include them into the review instead of excluding them due to different outcomes....
Response: Non-relevant outcomes refers to the independent of major research outcomes, they can not be combined using meta-analysis because of less than 2 studies, for example, original studies reported state anxiety, event-related Potential, integrated electromyogram, and subjective questionnaire (short-form of McGill pain questionnaire). Then we excluded them as the non-relevant outcomes in this current review.
5) The paper may benefit from a through English proofread as I have seen a few English errors,.
Response: The second author who is a native English speaker went through this revised manuscript accordingly.
6) Publication bias. The author consider the result of Egger's test p = 0.11 reflected no publication bias, which may not be correct, since p=0.11 reflect a high probability (89%) to imply some publication bias, but has not reach statistically significance level.
Response: Thank you for your response and insights. We have revised this sentence to indicate that there was no evidence of publication bias from a statistical significance standpoint.
7) Lifestyle issue. Both Qigong and Taichi is not designed as a clinical therapy, but a way of life in traditional Chinese medicine. The reviewed studies took TCE as a clinical therapy and show some significant clinical outcome. That is great! However, Taichi and Qigong is designed to generate a different lifestyle, which should affect the practitioner's behavior pattern and life-style in a long term. Future studies should keep that in mind, and include more quality of life and lifestyle outcomes in assessment.
Response: Thank you very much for your comments.
Reviewer 2 Report
Extensive review and meta analysis of tai chi and Qi Hong for low back pain. Conclusion is favorable but recommends large randomized control study.
The background and discussion could be shortened. Tables are ok. This reviewer would ask for a statistician’s review.
Author Response
Reviewer 2
Extensive review and meta analysis of tai chi and Qi Qong for low back pain. Conclusion is favorable but recommends large randomized control study.
Response: Thank you very much for your comments. As the number of RCT on this topic increases, we will conduct a follow-up review in the near future.
The background and discussion could be shortened. Tables are ok. This reviewer would ask for a statistician’s review.
Response: Thank you for your comments. Where appropriate, we have revised the introduction and background sections accordingly. Also, we have carefully reviewed the statistical analyses accordingly.